# Changes in Children and Youth’s Mental Health Presentations during COVID-19: A Study of Primary Care Practices in Northern Ontario, Canada

**DOI:** 10.3390/ijerph20166588

**Published:** 2023-08-17

**Authors:** Roya Daneshmand, Shreedhar Acharya, Barbara Zelek, Michael Cotterill, Brianne Wood

**Affiliations:** 1Thunder Bay Regional Health Research Institute, Thunder Bay, ON P7B 6V4, Canada; 2Section of Family Medicine, Northern Ontario School of Medicine (NOSM) University, Thunder Bay, ON P7B 5E1, Canada; sacharya@nosm.ca (S.A.); bzelek@nosm.ca (B.Z.); mcotterill@wawafht.com (M.C.)

**Keywords:** mental health, primary healthcare, youth, COVID-19

## Abstract

Recent research suggests that children and youth are at increased risk of anxiety and depression due to the indirect effects of the COVID-19 pandemic. In Canada, children and youths may face additional hurdles in accessing mental health services in rural areas due to socioeconomic disadvantages and healthcare provider shortages worsened by the pandemic. Our study aimed to assess changes in primary healthcare utilization related to depression and anxiety among children and youth aged 10–25 years in Northern Ontario, Canada. We analyzed de-identified electronic medical record data to assess primary healthcare visits and prescriptions for depression and anxiety among children and youth aged 10–25 years. We used provider billing data and reasons for visits and antidepressant/antianxiety prescriptions compared with 21 months pre-pandemic (1 June 2018 to 28 February 2020) and 21 months during the pandemic (1 April 2020 to 31 December 2021). Our interrupted time series analysis showed an average increase in visits by 2.52 per 10,000 person-months and in prescriptions by 6.69 per 10,000 person-months across all ages and sexes. Females aged 10 to 14 years were found to have the greatest relative change in visits across all age–sex groups. The greatest relative increases in antianxiety and antidepression prescriptions occurred among females and males aged 10 to 14 years, respectively. These findings indicate that there were increased anxiety and depression presentations in primary healthcare among children and youths living in northern and rural settings during the COVID-19 pandemic. The increased primary healthcare presentations of anxiety and depression by children and youths suggest that additional mental health resources should be allocated to northern rural primary healthcare to support the increased demand. Adequate mental health professionals, accessible services, and clinical recommendations tailored to northern rural populations and care settings are crucial.

## 1. Introduction

Globally, the COVID-19 pandemic has had direct and wide-ranging mental health impacts, including increased loneliness, anxiety, and depression in many different age groups due to lockdowns and the resulting isolation [1,2,3,4]. Additionally, indirect impacts have been reported due to diminished access to healthcare, including access specifically to counselors and healthcare professionals [5]. Studies highlight the greater risk of serious mental health conditions that are related to long-term stressors like the COVID-19 pandemic. in specific population subgroups, including children and youths [6,7]. For example, prolonged social isolation during these critical developmental years, in combination with other risk factors, including psychological and behavioral issues, can lead to fears, uncertainties, physical and social isolation, and extended school absenteeism [8]. These factors can have a negative impact, resulting in symptoms of anxiety, lethargy, impaired social interaction, and reduced appetite, which are associated with increased depressive symptoms [8,9,10,11]. Children and youths who face other systemic or structural barriers to care—such as those who reside in rural areas—likely face a higher risk for depression and anxiety symptoms [12,13,14]. 

Prior to the COVID-19 pandemic, Canadian children and youths experienced challenges in accessing appropriate mental health services, including low mental health literacy, fragmented care delivery, transitioning from youth to adult services, long waitlists, and a lack of culturally relevant services [15]. These challenges were exacerbated in rural and underserved areas where demographic, socioeconomic, and cultural determinants of health [16,17] also affect the availability and accessibility of children’s and youth’s mental health services. Poor socioeconomic conditions are prevalent in rural areas and have been correlated with increased levels of psychosocial stress, reduced physical activity, and limited access to preventive services [18]. Limited access to culturally relevant health services or experiencing racism can also exacerbate the challenges faced by rural residents in seeking mental healthcare, especially among Indigenous individuals and communities [19]. Children and youths from underserved areas were more impacted during COVID-19 by the loss of family income, the higher rates of morbidity and mortality among community members due to low-resource infrastructure, and problems in virtual connectivity [1]. Infrastructural and connectivity challenges disproportionally affect rural populations given the dramatic, now sustained, shift to virtual delivery of primary healthcare, including for mental health [2,20,21]. Furthermore, rural residents are less likely to access primary healthcare for their mental health concerns but more likely to present at an emergency department seeking acute care, suggesting they may not seek help until experiencing high levels of distress [22]. 

There is substantial research documenting the unintended mental health impacts of the COVID-19 pandemic on children and youths [23,24,25,26]. Unfortunately, underserved sub-populations, including northern rural populations, are not considered in most of the published research. Furthermore, a significant number of the current literature relies on self-reported data [23,27,28], which might contain measurement and sampling biases. These biases could inadvertently exclude these specific sub-populations from the analysis, limiting the comprehensiveness and accuracy of the findings. In Ontario, Canada, there is evidence of increased visits to pediatric emergency departments for mental health presentations, as well as admissions for suicidal ideation following the COVID-19 pandemic [29]. However, these analyses do not compare findings across geography or health system contexts. Without accurate information on how and to what extent the mental health of children and youths living in rural settings was impacted by COVID-19, funders, care providers, and decision-makers cannot appropriately tailor their care or shift resources to address a potentially significant health priority. 

In this study, we hypothesized that we would observe an increase in primary healthcare service utilization for depression and anxiety among children and youths in Northern Ontario during the COVID-19 pandemic when compared to the pre-COVID-19 period. Accordingly, our research question was focused on how and to what extent did the utilization of primary healthcare services for depression and anxiety among Northern Ontario children and youths (aged 10–25 years) change after the declaration of the COVID-19 pandemic? 

To address this knowledge gap, our team leveraged de-identified patient electronic medical record (EMR) data from a primary care practice-based research and learning network in Northern Ontario, Canada. Our secondary objective included assessing the frequency of prescriptions of antidepressant and antianxiety medications before and during the pandemic.

## 2. Materials and Methods

### 2.1. Study Design and Setting

We conducted a longitudinal retrospective cohort study of primary care utilization related to depression and anxiety among children and youths aged 10 to 25 years in a Northern Ontario practice-based research and learning network (NORTHH). NORTHH is the Northern Ontario network within the Canadian Primary Care Sentinel Surveillance network [30]. These networks are repositories of de-identified patient EMR data contributed by primary healthcare physicians and nurse practitioners. 

Northern Ontario, Canada, comprises over 90% of the province’s land mass but is home to only 6% of its population [31]. The population is generally older and has higher rates of chronic diseases and mental health conditions compared to the rest of the province. Northern Ontario’s population often faces significant barriers to accessing primary care providers in a timely manner [32,33]. Highlighted barriers to accessing healthcare in Northern Ontario include geographic isolation, workforce shortages, privacy concerns in small rural communities, differing health beliefs and attitudes compared to their urban counterparts, and a preference for receiving care from informal networks, including traditional providers [32]. The most recent Canadian Census estimates at least 15% of Northern Ontario respondents between 10 and 24 years of age identified as Indigenous [34], although this is likely an underestimate [35]. Indigenous populations living in Canada are more likely to experience poor health outcomes compared to non-Indigenous populations due to the impact of colonialism and systemic racism. Many Indigenous peoples live in rural and remote settings in Canada, including Northern Ontario, and disproportionally face challenges accessing culturally safe and appropriate healthcare services [36]. 

At the time of analysis, NORTHH had 60 family physicians and nurse practitioner providers who submitted patient EMR data to the network. Data extraction was facilitated by UTOPIAN (the University of Toronto Practice-Based Research and Learning Network), and the study received approval from the Ethics Committee at the University of Toronto, Office of Vice President, Research and Innovation (protocol code 23007, approved on 14 July 2021). The Lakehead University Research Ethics Board granted a waiver of ethics review for this study due to its secondary analysis of de-identified data.

### 2.2. Sample Population and Outcome Measures

Our sample population included children and youth patients aged 10 to 25 years who accessed primary healthcare with a NORTHH provider between 1 June 2018 and 31 December 2021. By extending the age range to include individuals up to 25 years of age, we aimed to capture the unique experiences and transitional effects that continue to occur during this important period. This age group represents a crucial stage where young people are still undergoing significant personal, social, and psychological changes [15,37]. We used two measures to describe primary healthcare utilization related to anxiety and depression: (1) the rate of unique patients with a diagnosis of anxiety or depression using the diagnostic billing code (ICD-9 codes 300, 309, 311, and 313); and (2) the rate of antidepressant/antianxiety medications prescribed by the primary care provider when the reason for the visit was either depression or anxiety. The list of medications considered in this analysis is listed in Appendix A. The rate of visits and rate of antidepressant/antianxiety prescriptions per 10,000 person-months were calculated by dividing the number of visits and the number of antidepressant/antianxiety prescriptions across the study duration (1 June 2018 to 31 December 2021), multiplied by 10,000. These measures have been used previously as part of algorithms designed to identify patients with a lifetime history of anxiety or depression [38,39,40]. 

### 2.3. Statistical Analysis

We first conducted a descriptive analysis of patient characteristics of the entire NORTHH cohort, looking at age and sex breakdown to understand the characteristics of our patient cohort (Appendix A). Then, we conducted univariate and bivariate analyses on the variables of interest, using frequencies and proportions to describe the absolute number of unique patient visits and prescriptions that occurred within the NORTHH cohort across the COVID-19 segments (i.e., pre-pandemic and during the pandemic). 

Our primary analysis compared (1) the rate of visits related to depression and anxiety per 10,000 person-months, and (2) the rate of antidepressant/antianxiety prescriptions per 10,000 person-months, before COVID-19 and during the COVID-19 pandemic. Rates were further stratified by age categories (10–14 years, 15–19 years, and 20–25 years) and sex (male, female). We used March 2020 as the COVID-19 “interruption” period because Ontario declared a state of emergency on 17 March 2020 [41]. This month divided the analytic interval into “pre-pandemic”, which includes the 21 months prior to the COVID-19 pandemic declaration (1 June 2018–28 February 2020) and the “during pandemic” interval of 21 months after the initial declaration (1 April 2020–31 December 2021). 

To understand longitudinal trends in a quasi-experimental (e.g., pre-post design), we used an interrupted time series [42] and segmented regression analysis to estimate the effects of the “interruption” (e.g., COVID-19 emergency) [43,44]. In the interrupted time series, data are collected consistently by equal intervals before and after an ”intervention” or “interruption” to examine the immediate and gradual effects of the intervention on the study outcome pre-and post-intervention [42]. With segmented regression analysis, we then calculated separate intercepts and slopes according to the two segments of pre- and during the COVID-19 pandemic. Segmented regression models fit the least squares regression line to each segment of the independent variable, time, and assumed a linear relationship between the rate of visits or rate of antidepressant/antianxiety prescriptions across time. The regression model was used to estimate the predicted rate of visits and predicted rate of prescriptions per 10,000 person-months, during the pre-pandemic segment and during the pandemic segment. We also adjusted for serial autocorrelation in the residuals, which arises because the observations taken over time are usually correlated [42]. Relative changes in rates were calculated using the predicted rates of the pre-pandemic and during-the-pandemic timelines. Data analysis was carried out using SAS software, Version 9.4. Copyright © 2013. SAS Institute Inc., Cary, NC, USA.

## 3. Results

### 3.1. Description of the Children and Youth Cohort and Mental Health Visits during the Study Timeline

At the time of analysis, there were 23,378 unique patients in NORTHH, with approximately equal amounts of males and females (Appendix A). Thirty-four percent of the NORTHH cohort were children and youths aged 10 to 25 years. Of the 3974 children and youths between 10 and 25 years, 2048 patients were males, and 1926 were female. Figure 1 and Table 1 and Table 2 present the absolute number of primary care visits and medication prescriptions prior to the pandemic and during the pandemic, respectively. 

### 3.2. Segmented Regression Models

According to residual diagnostics and model fit characteristics, the regression models used to predict the expected visit and prescription rates pre- and during the pandemic were a good fit for our data set. The baseline rate of primary care visits related to anxiety and depression on 1 June 2018 was 75.1 per 10,000 person-month. In the pre-pandemic segment (1 June 2018 to 28 February 2020), the average monthly rate of visits appeared relatively stable, only decreasing by 0.13 per 10,000 person-month. In March 2020, the average rate of primary care visits showed a decrease of 7.2 per 10,000 person-month. From April 2020 forward, the predicted rate of primary care visits due to anxiety and depression increased at an average rate of 2.5 per 10,000 person-month (95% CI: 0.76–3.69). 

The baseline rate of prescriptions related to anxiety and depression on 1 June 2018, was 99.2 per 10,000 person-month. In the pre-pandemic segment, the average monthly rate of prescriptions increased by 58.8 per 10,000 person-month. In March 2020, the predicted rate of prescriptions decreased by 66.0 per 10,000 person-month. From April 2020 and forward, the predicted rate of prescriptions related to anxiety and depression increased at an average rate of 6.7 per 10,000 person-month (95% CI: 1.9–9.5). 

#### 3.2.1. Comparing the Predicted Rate of Primary Care Visits per Person per Month for Anxiety and Depression Pre- and during the COVID-19 Pandemic

During the entire study period, 773 patients (males: 60%, females: 40%) had 3675 primary care visits related to anxiety/depression, of which 1697 visits happened in the pre-pandemic timeline and 1978 visits happened during the pandemic timeline (Figure 1 and Table 1). 

The predicted rate of primary care visits for anxiety or depression showed an increase of 45.5 visits per 10,000 person-month after April 2020, compared to the pre-pandemic segment. This reflects a relative change of 63% (Appendix A). When stratifying by sex, females had a higher increase in the predicted rate of visits during the pandemic timeline compared to males (125.5 visits per 10,000 person-month vs. 107.8 visits per 10,000 person-month). This comparison reflects a relative change of 65% in the predicted rate of primary care visits among females compared to 50% among males (Appendix A). The predicted rate of visits increased after the pandemic declaration in all age groups. Age–sex-stratified analyses showed increases in the predicted rates of visits after April 2020 across all sub-groups, except for males aged 10–14 years and females aged 20–25 years. A decrease was observed in the predicted rate of visits after the pandemic was declared in both of these sub-groups. Male children aged 10–14 years had the highest decrease in the predicted rate of primary care visits compared to the other sex and age groups (a relative change of −59% after the pandemic was declared), while female children aged 10–14 years had the highest increase in the predicted rate of primary care visits compared to the other age-sex stratified sub-groups (a relative change of 209%) (Appendix A).

#### 3.2.2. Comparing the Predicted Rate of Antidepressant/Antianxiety Prescriptions per Person per Month Pre- and during the COVID-19 Pandemic

There were a total of 3885 antidepressant/antianxiety prescriptions during the study period from 1 June 2018 to 31 December 2021 among NORTHH patients aged 10 to 25 years. Of these, 40% (n = 1324) of antidepressant/antianxiety prescriptions were prescribed prior to 1 March 2020, and 60% (n = 2561) were prescribed between 1 April 2020 and 31 December 2021 (Figure 1 and Table 2). The average monthly rate of relevant prescriptions was slightly higher than the average monthly rate of primary care visits during the pandemic timeline. The predicted rate of antidepressant/antianxiety prescriptions across all children and youths in NORTHH increased by 74.5 per 10,000 person-months during the pandemic segment compared to the pre-pandemic segment (a relative change of 33.5%) (Appendix A). Comparing males and females, females had higher predicted rates of antidepressant/antianxiety prescriptions during the pandemic compared to males (106.4 antidepressant/antianxiety prescriptions per 10,000 person-month vs. 23.6 antidepressant/antianxiety prescriptions per 10,000 person-month, respectively). This reflects a 50% relative increase in prescriptions among females compared to 10% relative change in males (Appendix A). In the age-sex stratified analysis, youth in the 20–25 years age group showed a decrease in the predicted rate of antidepressant/antianxiety prescription both in males and females, while the highest increase was in children and youths aged 10–14 years, particularly in males (Appendix A).

## 4. Discussion

This study represents the first analysis of primary healthcare EMR data for anxiety and depression among children and youths aged 10–25 years in northern rural settings in Ontario, Canada. Using an interrupted time series analysis, we observed significant increases in predicted rates of primary care visits and antidepressant/antianxiety prescriptions during the COVID-19 pandemic compared to predicted pre-pandemic rates. The relative increase in the predicted rate of primary care visits was almost twice as high as the relative increase in the rate of medication prescriptions (63% vs. 34% per 10,000 person-months). Females had higher predicted rates of primary care visits and prescriptions during the pandemic compared to males (relative change of 65% vs. 50% for primary care visits and 50% vs. 10% for medication prescriptions), even though this was not true prior to the pandemic. When we examined trends according to age–sex subgroups, the greatest increase in the predicted rate of visits occurred among female children and youths aged 10–14 years. Children and youths aged 10–14 years, especially males, had the highest increase in the predicted rate of antidepressant/antianxiety prescriptions after March 2020, while youth aged 20–25 years showed decreases in the predicted rate of prescriptions, across both sexes.

A provincial report suggests that Northern Ontario children and youths consistently report worse mental health outcomes compared to provincial counterparts, largely attributable to unique determinants of health of rural and northern communities [45]. Factors that are correlated with poor mental health outcomes in Northern Ontario include relative isolation, limited transportation options, shortage of mental health professionals and specialist services due to a suboptimal distribution of providers, long wait times for physicians, and long distances traveled for referrals [46]. Unique determinants of health in northern and rural communities include lower socioeconomic status, higher rates of poverty, limited educational opportunities, and higher rates of unemployment [47]. Individuals living in northern rural communities may feel stigmatized if they seek help for their mental health concerns [48]. Poorer outcomes are also reported in adult populations in Northern Ontario, such that residents report higher rates of depression and use more prescription medications for sleep, anxiety, and depression [16]. Additionally, the hospitalization rate for mental health concerns, particularly suicide-related hospitalizations, is twice as high in Northern Ontario compared to the rest of Ontario [16]. Although our study does not compare results to primary healthcare practices outside of Northern Ontario, our findings confirm a comparative geographic analysis of mental health service utilization is warranted. We observed a significant increase in care utilization by children and youths after the COVID-19 pandemic was declared, yet further investigation is required to assess how outcomes changed among children and youths. 

Over the past decade, mental health issues have become more prevalent among children and young people seeking healthcare, particularly among female patients [49]. Phillips et al. [49] found an increase in primary healthcare presentations for depression and anxiety in females in Ontario aged 10–14 years and 15–19 years between 2012 and 2017. Then, global estimates during the first year of the COVID-19 pandemic indicated a significant increase in child and youth mental illness [6]. Stephenson et al. [26] found initial decreases in anxiety/depression-related visits during the pandemic surpassed pre-pandemic levels by the end of 2020, using primary care EMR data from a large urban population in Ontario, Canada. Our findings illustrated a similar trend that there was an initial sharp decrease in primary healthcare utilization in March 2020. In the first few months after March 2020, patients began to access healthcare services more frequently, particularly with an increase in virtual visits [50]. In our study, we found that by the end of 2021, children and youths were more likely to have anxiety or depression-related primary care visits or receive an antidepressant/antianxiety prescription from their primary care provider than before the pandemic. These findings contribute to the growing evidence of increased demands for mental health services in this age group [6,26,51,52], suggesting a potentially important impact on anxiety and depression population health outcomes. 

Our data aligns with the well-established findings that women generally have higher rates of depression and anxiety compared to men, including during the pandemic [53,54,55]. Some research has suggested that girls and women have poorer outcomes when exposed to stressors and uncertainty, like the COVID-19 pandemic [56]. Hormonal fluctuations during puberty and social behaviors may contribute to their susceptibility to stressors [57,58]. Studies show that by the age of 15, females are around twice as likely as males to have experienced at least one episode of depression, and this disparity continues to persist for the next 35 to 40 years. We found a significant increase in the rate of visits to primary care providers by female patients following the declaration of the pandemic, with the most pronounced relative increase observed in females aged 15–19 years. The higher predicted rates of visits and prescriptions related to anxiety and depression among females could be related to gendered prescribing practices of providers, such that girls and women are at greater risk of “overtreatment” of mental health conditions [59,60]. EMR data does not often reflect the complexities of sex and gender [61], so our analysis does not capture the complexity of these constructs beyond a single binary that typically reflects sex at birth. 

Although the overall prescription rate was higher for females, both males and females in the 10–14-year age group experienced the most significant increase in prescription rates. However, this might be an exaggerated finding because of the low number of prescriptions prior to the pandemic. Our findings differ from those of Stephenson et al. [26], who did not observe an increase in antidepressant/antianxiety prescriptions despite an increase in primary care visits within the urban study sample. Primary care providers in our study sample might have been more likely to prescribe antidepressant/antianxiety medications because there are limited options for alternative treatments, such as publicly funded counseling services. The observed increase in antidepressant/antianxiety prescriptions might also be related to changes in providers’ behavior and tendency to prescribe medication compared to other, possibly more appropriate, treatment modalities for depression and anxiety, such as psychotherapy or cognitive behavioral therapy. Also, due to the limited mental health services for children and youths in northern and rural communities, patients in our study sample might have presented with more severe symptoms and, consequently, received prescriptions for anxiety or depression medications. We found that primary care visits and prescriptions for anxiety and depression were lowest among children and youths aged 20–25 years; this may be due to their focus on education or work, resulting in greater self-reliance and less disruption to their routines during the pandemic. Additionally, older individuals may have developed better coping mechanisms and resilience, leading to lower rates of seeking mental healthcare compared to younger individuals [62].

We acknowledge some limitations of our study. Our analysis did not include data from non-physician primary care encounters, which means the findings will not represent the utilization of mental health services from interprofessional and community-based healthcare providers during the pandemic. Many rural practices included in the study rely on interprofessional teams of nurse practitioners, physician assistants, and allied health professionals to provide a mental health diagnosis, counseling, and/or treatment. Patient encounters with these interprofessional providers are not consistently recorded in billing data. As a result, our study might have underestimated the rates of depression and anxiety presentations among children and youths as well as prescription rates. Therefore, when interpreting the provider billing data from EMRs as a proxy for the patient’s health condition, we must account for the alternative payment models (i.e., not fee-for-service) and practice patterns by the providers and practices in our Northern Ontario sample. Additionally, EMR data entry practices vary by provider and by practice, so our analysis that relies heavily on billing codes and the reason for the visit as a proxy for health conditions may oversimplify the heterogeneity in health concerns among our cohort. Therefore, we consider the findings of this study important for hypothesis generation and streamlining further investigations. 

Our study also did not incorporate data from hospital emergency rooms, which serve as the initial point of contact for many patients in northern rural Ontario, given the limited health workforce and infrastructure within these settings. Saunders et al. [63] reported an increase in visits to pediatric hospitals for mental health presentations during the COVID-19 pandemic, alongside a simultaneous decrease in presentations to non-pediatric hospitals and emergency departments. Primary healthcare practices included in our study sample heavily rely on family physicians who work in both emergency departments and clinics. Therefore, the observed increase in primary healthcare visits may reflect improved access to primary care settings in Northern Ontario compared to seeking care in emergency departments. 

## 5. Conclusions

In conclusion, our study of Northern Ontario primary care practices shows significant increases in primary health care visits and antidepressant/antianxiety prescriptions among children and youths aged 10 to 25 years during the COVID-19 pandemic. We found that female children and youths had higher predicted rates of primary care visits and prescriptions during the pandemic compared to males, although this was not true prior to the pandemic. When comparing our findings to the current evidence on primary care utilization for anxiety and depression, we found significantly increased rates of prescriptions for antidepressant/antianxiety medications among youths after the pandemic declaration. These findings highlight an important population health need within northern rural settings and suggest that primary care encounters might involve more severe presentations of anxiety or depression. Acknowledging that children and youths have an increased demand for primary mental healthcare suggests that additional resources should be allocated to the mental health workforce (e.g., allied health professionals, counselors, therapists), and primary or community health programming specifically for these age groups, particularly for populations living in northern rural settings. Further investigations into age–sex differences in primary care utilization and prescriptions for anxiety and depression within children and youths will be important for tailoring program design and improving how and what primary care is delivered. We recommend that future studies apply a geographic or contextual analysis to better understand how and to what extent mental health services and population health outcomes vary across settings. 

## Figures and Tables

**Figure 1 ijerph-20-06588-f001:**
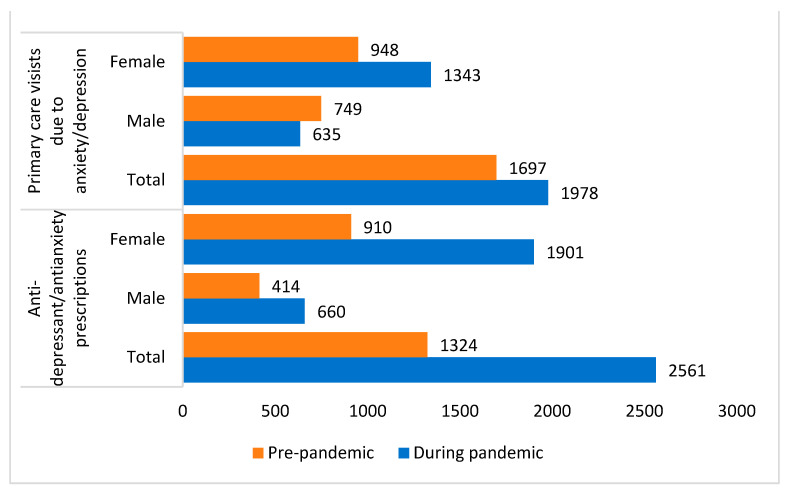
Total number of primary care visits due to anxiety/depression and antidepressant/antianxiety prescriptions among NORTHH children and youth patients.

**Table 1 ijerph-20-06588-t001:** Primary care visits due to anxiety/depression among NORTHH child and youth patients.

	Pre-Pandemic (1 June 2018–28 February 2020)			During Pandemic (1 April 2020–31 December 2021)		
	Total, N (%)	Male, N (%)	Female, N (%)	Total, N (%)	Male, N (%)	Female, N (%)
10–14 years	185 (10.9)	91 (12.1)	94 (9.9)	146 (7.4)	58 (9.1)	88 (6.5)
15–19 years	577 (34)	262 (35.0)	315 (33.2)	870 (44.0)	202 (31.8)	668 (49.7)
20–25 years	935 (55.1)	396 (52.9)	539 (56.9)	962 (48.6)	375 (59.1)	587 (43.7)

**Table 2 ijerph-20-06588-t002:** Received at least one antidepressant/antianxiety prescription among NORTHH child and youth patients.

	Pre-Pandemic (1 June 2018–28 February 2020)			During Pandemic (1 April 2020–31 December 2021)		
	Total, N (%)	Male, N (%)	Female, N (%)	Total, N (%)	Male, N (%)	Female, N (%)
10–14 years	21 (1.6)	7 (1.7)	14 (1.5)	99 (3.9)	21 (3.2)	78 (4.1)
15–19 years	319 (24.0)	86 (20.8)	233 (25.6)	1019 (39.8)	225 (34.1)	794 (41.8)
20–25 years	984 (74.1)	321 (77.5)	663 (72.8)	1443 (56.3)	414 (62.7)	1029 (54.1)

## Data Availability

The data are not publicly available due to ethical and healthcare privacy concerns.

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
