# Peer review of "Changes in Children and Youth’s Mental Health Presentations during COVID-19: A Study of Primary Care Practices in Northern Ontario, Canada"

_ijerph, 2023, doi:10.3390/ijerph20166588_

Round 1

Reviewer 1 Report

Reviewer general comment:

The manuscript by Roya Daneshmand and colleagues examines changes in primary health care utilization related to mental health, depression and anxiety, by children and youth in a rural area of Canada, comparing pre and during COVID-19 pandemic periods. The manuscript is well-organized, clear, and concise, making it relevant to the field. The study design is adequate. The manuscript provides enough relevant information for researchers, professionals, and clinicians working in mental health. The references are appropriate, being up to date, and the authors avoid excessive self-citations. I have only few comments and suggestions which are listed below, in no particular order.

Comments and suggestions:

Introduction, methods, and results:

1. Please, describe risk factors (line 43, reference 8).

2. Lines 96-97: further describe the significant barriers that the groups often face accessing primary care providers in a timely manner.

3. Table S2 lacks clarity regarding “income quintile”, and this aspect is not addressed in the methods, results, or discussion sections. Please provide clarification.

4. Table S1 does not offer descriptive results, please review.

5. Please review the disparity between the number of females stated in line 164 and the corresponding value presented in Table S2.

6. I suggest using one decimal.

7. Information on sociodemographic factors (e.g., education level, self-identification as indigenous), risk factors (e.g., substance use/disorders), and clinical characteristics (e.g., other mental health conditions) of the population should be included, as adjustments, to enhance result clarity and to account for potential confounders in the model. This should also be incorporated to improve the discussion.

8. Data about history of COVID-19 infection in the different groups could provide interesting information to the tables.

9. Exploring the correlations between demographic characteristics and primary care visits and/or prescriptions for depression and anxiety would offer valuable insights.

10. It would be good to know the rates dividing the COVID-19 pandemic period in different sub-periods (set points), considering at least 3 different COVID waves in order to comprehend the impacts of varying potential environmental stressors.

11. Please provide additional information about the group that received prescript medication during the pandemic. For example, how was their mental health condition prior to the pandemic?

12. I suggest to be cautious when interpreting the results.

Discussion:

1. Please provide an expanded explanation in the second paragraph (reference 44), and supplement it with additional references.

2. The limitations of this study are addressed properly.

3. The limitation mentioned in lines 309-312 resembles the one described in lines 324-329. Simplifying it is recommended.

4. Lines 154-156 are not completely clear.

Reviewer 2 Report

Dear Authors,

The present research article, entitled “Changes in Children and Youth Mental Health Presentations During COVID-19: A Study of Primary Care Practices in Northern Ontario, Canada”, aims to determine changes in primary health care utilization related to depression and anxiety in children and youth aged 10-25 in Northern Ontario, Canada.

The main strength of this manuscript is that it addresses a relevant and timely issue, reporting an increase in presentations of anxiety and depression in primary health care settings among children and youth living in rural and northern settings, during the COVID-19 pandemic, populations that tend to be less studied.

In general, I believe that the topic and approach of this article is timely and of interest to the readers of Brain Sciences. However, I believe that some issues should be included to improve the quality of the manuscript.

1. The authors are invited to complete the research hypotheses and expected outcomes of their research.

2. The data in the 1a and 1b tables could partly be presented in the form of a graph.

3. Another important aspect to highlight would be to explain how the diagnoses of anxiety and depression are determined. This is especially relevant because, on many occasions, primary care professionals do not follow a protocol or are not trained to make such a diagnosis, which can lead to erroneous conclusions. Consequently, there may be an over-prescription of antidepressants/anxiolytics or a misuse of them.

4. In line with the previous suggestion and based on the results found, it is essential to place greater emphasis on the importance of the existence of specific mental health resources in rural areas.

I believe that the manuscript may carry important value in studying in primary health care utilization related to depression and anxiety by children and youth 15 aged 10-25 years.

 Best regards,

Reviewer 3 Report

This paper provides an interesting analysis of electronic medical records in order to understand the impact of the COVID-19 pandemic on the mental health of children and young people living in remote locations in Ontario, Canada. An extremely topical subject.

I found that the paper is well written, its rationale, results and conclusions easy to follow. I would only suggest that the authors add a little bit more about how their findings fit in the global context. For example, are these findings likely to be similar elsewhere, and how could practitioners and researchers in other countries learn from this study, not only in terms of its findings, but also in terms of the possibility of carrying out comparable studies. 

The next question to the one addressed on the paper, would be around how persistent the increase in mental health services utilization has been since the end of the pandemic has been officially declared. In other words, has demand remained high? Possibly not enough time has elapsed to allow such exploration at present, but it is something the authors might want to add to their recommendations for future research.

Minor edits

·       ln 20 – specify the timelines (i. e. April 2020 – December 2021) instead of “after March 2020).

·       ln. 35-37 – The first sentence in this paragraph is somewhat hard to follow. Consider “unpacking” in two sentences, one focusing on the general effects, and the other on what that looks like for young people. Avoid the repetition of “including” and of “impacts” in short succession.

·       ln. 39-41 – Consider revising this sentence to read “... serious mental health conditions that are related to long-term stressors like the COVID-19 pandemic in specific population subgroups, including children and youth.”

·       ln. 43 – Replace “are” with “is”, or use “and” instead of “in combination with”

·       ln. 95 – Replace “have” with “has” (to read “the population is generally older and has...”

·       ln. 96 – Referring to “People” or “Residents” instead of “They often...” will make the sentence easier to follow.

·       ln. 98 – Briefly explain the impact of identifying as Indigenous for readers who may be less familiar with the Canadian context / or whose experience is in less diverse settings.

·       ln. 107 – “n” is missing in “sample population”

·       ln. 108 – Replace “youth patients” with “young patients”.

·       ln. 125-126 – This sentence is incomplete: “to understand how our patient cohort”...what? 

·       ln. 146-147 – “pre” and “post” sometimes are hyphenated sometimes not. 

·       ln. 231 – add “in Canada” after “rural settings”.

·       ln. 241-243 – if the differences were not statistically significant, then there are no differences. Please revise the wording to reflect any ambiguity inherent in the results (e. g. results not statistically significant, but otherwise reason to believe there may be an effect that failed to be detected).

·       ln. 323 – move “(i. e., not fee-for-service)” to appear after “models”.

The paper is well written and generally easy to follow. It could benefit from a few, but very minor, edits, as suggested in the comments and suggestions for authors.

Round 2

Reviewer 1 Report

Dear authors,

Thank you for your effort in refining the manuscript. The article now stands as a comprehensive and well-structured contribution to the field.

Reviewer 2 Report

Dear authors,

Thank you for your effort in refining the manuscript. The article now stands as a comprehensive and well-structured contribution to the field.

Kind regards.